# Prevalence and characterization of asymptomatic thyroid nodules in Assin North District, Ghana

**Martin Tangnaa Morna** [1‡]*, **Derek Anamaale Tuoyire**[2◦], **Bashiru Babatunde Jimah**[3◦], **Sebastian Eliason**[2], **Anthony Baffour Appiah**[1,4◦], **Ganiyu Adebisi Rahman**[1‡]

1 Department of Surgery, School of Medical Sciences, University of Cape Coast, Cape Coast, Ghana,
2 Department of Community Medicine, School of Medical Sciences, University of Cape Coast, Cape Coast, Ghana, 3 Department of Medical Imaging, School of Medical Sciences, University of Cape Coast, Cape Coast, Ghana, 4 Ghana Field Epidemiology and Laboratory Training Programme (GFELTP), School of Public Health, University of Ghana, Accra, Ghana

◦ These authors contributed equally to this work.
‡ These authors also contributed equally to this work.
* drmorna@yahoo.com

**Data Availability Statement:** Data are available from the Zenodo database (https://doi.org/10.5281/zenodo.5840804).

## Abstract

### Background

Ultrasound is now the initial imaging modality of choice for detection and characterization of lesions of the thyroid gland. Ultrasound imaging studies of the thyroid gland report varied prevalence of asymptomatic thyroid nodules (ATN), ranging from 20 to 67%. This study estimated the prevalence, characterized and determined factors associated with ATN in selected communities in the Assin North Municipality, Central Region, Ghana.

### Methods

The study was a cross-sectional design, involving 320 participants from six (6) communities in the Assin North District of the Central Region of Ghana. Socio-demographic data and data from ultrasound examination of the thyroid gland were analyzed using descriptive and inferential statistical techniques.

### Results

The prevalence of ATN was 11.3% among 320 participants with the mean age of 56.53 (±16.5) years. ATNs were common in the left lobe (69%) and predominantly solitary (64%). ATNs increased with age and body mass index (BMI). Those aged 60 years and above had significantly higher odds (OR = 24.40, 95% CI = 2.59–229.86) of having ATNs, likewise overweigh (OR = 5.32, 95% CI = 1.12–25.20) and obese (OR = 12.51, 95% CI = 1.47–106.58) individuals.

### Conclusion

The prevalence of ATN in our study population was relatively low, and more predictable among those 60 years or older, those in unhealthy BMI categories. There is the need for the

**Funding:** The authors received no specific funding for this work.

**Competing interests:** The authors have declared that no competing interests exist.

reinforcement and intensification of educational campaigns on the consumption of iodized dietary salt as well as the consumption of foods rich in iodine content, particularly among older individuals.

## Introduction

The thyroid gland is a butterfly shaped endocrine gland located superficially in the infrahyoid compartment of the neck, within the space laterally outlined by muscles anteriorly, trachea and esophagus posteriorly, carotid arteries and jugular veins [1, 2]. It functions in the production, storage and secretion of thyroid hormones thyroxine (T4) and triiodothyronine (T3) [2–4]. These hormones play essential physiological roles in the body including regulation of body metabolism and ensures growth and development of the individual [1, 3]. The size and shape of the gland varies widely in normal individuals and these anatomical and physiological differences exist across continents and ethnic groups [1]. Although the physiological and pathological determinants of the thyroid volume have been well established, there is growing evidence of the influence of factors such as age, sex, body mass index, body surface area, iodine among others on thyroid volume [5–9].

A common pathology associated with thyroid gland is the development of thyroid nodules, which is essentially an abnormal growth of the cells of the thyroid gland. Early stages of thyroid nodules are mostly asymptomatic and undetectable by physical examination. As such they are only often detected during routine neck imaging at which point they may be classified as incidental thyroid nodules [2]. Nonetheless, the timely diagnosis of these asymptomatic thyroid nodules (ATN) presents an opportunity for early interventional therapy to avert complications including hyperthyroidism and goiters. The advent of advanced imaging modalities (CT, MRI, and ultrasound) has significantly improved the diagnosis of ATN, as well as research studies on the subject.

Available studies on ATN show variations in both prevalence and associated factors. For instance, a multi-center study [10] of ATN in Asia reported a prevalence of 34% and 12.5% for ATN and pure cysts, respectively. One study in Africa based on ultrasound imaging reported a prevalence of thyroid incidentalomas of 22.4% [11], while a number of studies elsewhere suggest an increasing trend of cancerous thyroid nodules [7, 8, 11–13]. The pattern generally shows higher prevalence of ATN in females than in males and, with advancing age [5]. Mohammed et al. [14] found women to have 2.8 times odds of developing ATN compared to men. Other factor found to be associated with ATN include body mass index (BMI), waist circumference, and body fat composition [10, 15, 16].

Thyroid disorders contribute significantly to the burden of diseases in Ghana [17–20]. A few studies have highlighted the increasing trend of thyroid disorders in two major cities in Ghana, Greater Accra [19, 20] and Greater Kumasi [17]. A hospital-based retrospective study by Sarfo-Kantako et al. [17] conclude that the prevalence thyroid disorders in Ghana remains high despite over 20 years educational campaigns on the consumption of iodized salt that aimed at preventing incidence of thyroid disorders [17, 18]. Their study revealed that nearly half (48%) of all thyroid admissions are from multinodular goiters (22.5% toxic and 25.5% nontoxic), followed by hypothyroidism (13.1%), diffuse toxic goiter (12.1%), nontoxic diffuse goiter (6.6%), and toxic adenoma (2.1%) [17]. Moreover, these studies are hospital-centered with cases reported from different region which do not represent the various populations patients' reside. Also, most of these patients report at advanced stage with total thyroidectomy

as the only effective option for more than 40 percent of them [19, 20]. To inform early detection and timely treatment in limited-resource setting like Ghana, there is the need to shift from routine clinical practices in hospitals (hospital-entered approach) and reach out to the population (community-entered approach). Reliable primary data is required in Ghanaian setting to fill this knowledge gap on the subject and promptly inform medical practices locally. Community-centered screening survey allows somewhat timely identification of both symptomatic and asymptomatic people living with thyroid disorders, provide health education to cause behavioral changes while leading them to well-equipped health facilities for care. This study in the Assin North District estimated the prevalence, characterized and determined factors associated with ATN in six communities. It also provided us the opportunity to early detect people with thyroid nodules and arrange them for proper medical and surgical care. Evidence from this study should inform a broad assessment of thyroid glands among Ghanaian population which will estimate the true incident of thyroid disorders as well as assessing the Ghana iodize salt consumption campaign.

## Materials and methods

### Study design and sampling

The study was cross-sectional involving six (6) communities in the Assin North District of the Central Region of Ghana. The communities (Bremang, Dense, Aboteriyie, Ahuntem, Achiano, and Kushea) have been adopted by the University of Cape Coast, School of Medical Sciences (UCCSMS) as a social laboratories for its Community Based Experience and Service (COBES) programme. COBES is a flagship programme whereby students as part of their medical training spend three to four weeks each academic year in selected communities interacting and empirically studying the population health dynamics in these communities. Thus, the study was conceived and tailored into the COBES programme of 2019. Ethical approval was obtained from the Institutional Review Board of the University of Cape Coast, Ghana, with this reference number: UCCIRB/EXT/2017/18. All community entry protocols with local authorities were observed before the study commenced. Further, protocol involving informed consent was also duly observed during and after this study. Both verbal and written consent were obtained from each participant prior to participation. Participation in the study was strictly on voluntary basis.

The sample size for the study was determined using the Cochran's formulae for estimating the sample size for a large population, n = $Z^2 [p(1-p)]/e^2$ where n = minimum sample size, Z = value from the standard normal distribution of a specified confidence level, $e$ = margin of error and $\sigma$ is the population standard deviation. Hence, minimum sample size of 311 was estimated using the Africa average (25%) prevalence of asymptomatic thyroid nodule [11, 21], margin of error of 0.05 and at 95% confidence interval given us Z = 1.96. Factoring in a 10% non-response, a final sample size of 343 was arrived at.

Distributing this estimated sample size of 343 based on proportion-to-population size, the sample size allocation for each of the six (6) communities was as follows: Kushea = 219; Bremang = 25; Aboteriyie = 18; Ahuntem = 24; Dense = 15 and; Achiano = 42.

Each of these six (6) communities was considered a stratum in which all households were listed to constitute a sampling frame from which the respective number of households in each community were sampled using systematic random sampling technique. The sampling interval ($K^{th}$) in each community was determined by dividing the total number of listed households (N) by the number of households respectively require (n) (based on proportion-to-population size) as shown in Table 1. The simple random sampling technique was then employed to select the first household ($i<K$) in each community, from which every $K^{th}$ household was selected

**Table 1.  Distribution of samples selected from six communities.**

| Community | Population | No. households | Sampling interval (X) | Sampled households | Individuals sampled |
|---|---|---|---|---|---|
| Bremang | 701 | 175 | 7 | 25 | 25 |
| Aboteriyie | 514 | 99 | 6 | 18 | 18 |
| Dense | 396 | 58 | 4 | 15 | 15 |
| Ahuntem | 669 | 167 | 7 | 24 | 24 |
| Achiano | 1170 | 293 | 7 | 42 | 42 |
| Kushea | 6126 | 1502 | 7 | 219 | 219 |
| Total | 9576 | 2394 | 7 | 343 | 343 |

until the expected number of households in each community was met. One eligible consenting participant in each selected household was then randomly selected for the study. In a few instances where there were no consenting or eligible participant, the next household on the roll was considered.

Exclusion criteria included participants with anterior neck swelling or clinical evidence of thyroid disease, smokers, persons on lithium, phenytoin, oral contraceptive drugs, and women during menstruation, pregnant women or women who had delivered within the last 12 months and persons with any systemic disorder.

## Data collection

Data collection was conducted in July, 2019 in two phases. The first phase consisted of face-to-face interviews with participants using a structured interview guide to elicit socio-demographic information such as age, sex, marital status, and highest level of education; history dietary salt intake (often intake- at least 5g or one teaspoon of iodized salt per day and not often intake- less than 5g or one teaspoon of iodized salt per day or not at all); and history of alcohol intake (yes- consumed alcohol regardless of quantity and no- had not consume alcohol before). Anthropometric measurements of body weight (kg) and height (cm) were measured using standard anthropometric techniques and further computed to generate measures of body surface area (BSA) and body mass index (BMI). Data collection in the phase was conducted by six (6) trained research assistants from UCCSMS and duly supervised by key investigators (listed authors) of the study.

The second phase of the data collection mainly focused on diagnostic imaging of the thyroid gland by a specialist radiologist with over five (5) working experience in thyroid examination using various imaging technologies. A screening center was staged in each of the study communities on different days while ensuring that such days did not conflict with market days or other important community events. Participants who were interviewed at the household level were given an identification chit to present with to the screening stage for easy synchronization of their interview data with thyroid data. Given that ultrasound has been recognized as the initial imaging modality of choice for the early detection of thyroid nodules [2, 9, 22, 23], a real-time ultrasound scanner (MEDISON SA8000SE-MAI, 1003 Dachi-Dong, Gangnam-Gu, Seoul Korea) with a 7.5 MHz, 50 mm linear transducer was used in examining the thyroid gland of study participants.

Participants were examined while in a supine position with hyperextended cervical spine. Ultrasound gel was applied over the thyroid area with the transducer directly placed on the skin over the thyroid gland. Longitudinal and transverse scans were performed, to obtain length and width in centimeters, of each thyroid nodule. If there were multiple nodules in a single thyroid lobe only the dimensions of the largest were recorded. Documented

characteristics of thyroid nodules included the location of nodules in the thyroid lobe; number of nodules (solitary or multiple), nodule composition (cyst, solid or mixed), calcifications, and nodule size (length and width in centimeters). Out of the 343 participants interviewed in the initial phase of the data collection, 23 participants failed showed up for the thyroid screening in the second phase despite countless attempts to contact them. Hence the current study is based on 320 participants who were successfully interviewed and screened.

## Statistical analysis

The data was captured using SPSS and later exported to STATA 11.0 for further management and analysis. A protocol was designed from the outset for imputing, ensuring data quality and preserving of data for reuse. Descriptive statistics including frequencies, percentages, means and standard deviation were used to summarize participants' socio-demographic and thyroid characteristics. Bivariate and multivariate logistic regression analyses were conducted to determine factors associated with ATN. Odds ratios and corresponding confidence intervals were reported with statistical significance at $p < 0.05$.

## Results

As presented in Table 2, the study involved 195 women and 125 men with a mean age of 56.5 (±16.5). Over nine in ten participants often (97%) consume dietary salt, while just a quarter (25%) of them take in alcohol. With respect to anthropometry, the participants were generally in the healthy category of BMI with a mean of 23 (±5), although about 28% of them were classified as overweight or obese. BSA was averagely 1.6 (±0.3).

From the ultrasound examination of the thyroid gland of participants, the prevalence of ATN was estimated to be about 11% (n = 35). Sex, age, marital status, alcohol intake and BMI were found to be significantly associated with the development of ATN. Further, the prevalence of ATN seems to increase with age (22% among those aged 60 year or more) and BMI (32% for in the obese category), but decreased level of education increased ATN. Regarding marital status ATN was more common among those formerly married (27%).

The distribution of asymptomatic thyroid nodules by community are summarized in Fig 1. A significant proportion of participants at Achiano (15.6%), Kushea (14.9%) and Ahuntem (10.3%) were diagnosed of thyroid nodules.

The characteristics of ATN diagnosed with ultrasound in the study are presented in Table 3. About 64% were solitary while 36% were multimodula. A greater proportion of the nodules were in the left lobe (39%) with 11% of nodules found in multiple locations (both lobes and isthmus). A minimal occurrence of cyst (6%) and calcification (3%) in the nodules diagnosed were recorded. The average length and width of the nodules were 1.2 (±0.5) and 0.9 (±0.5), respectively.

The results of the logistic regression analysis are presented in Table 4. The bivariate model does not differ from the multivariate model in terms of statistical significance and direction of effect, except for increased odds in the multivariate model. In addition, the positive significant association between sex and the development of ATN in the bivariate model was not sustained in the multivariate analysis. Overall, the results of the logistic regression reveal that the odds of developing ATN were significantly higher (OR = 24.40, 95% CI = 2.59–229.86) for those aged 60 years or older compared those younger than 20 years old. BMI was also found to be significantly associated with the occurrence of ATN with the highest odds among those with in the obese (BMI ≥30) category (OR = 12.51, 95% CI = 1.47–106.58) compared with the reference category (BMI<25).

**Table 2. Sociodemographic and anthropometric characteristics of study population.**

| | Total subjects | Number nodule (proportion) | P-value |
|---|---|---|---|
| **Sex** | | | 0.001[a] |
| Male | 125 (39.1) | 5 (4.0) | |
| Female | 195 (60.9) | 31 (15.9) | |
| **Age (years)** | | | 0.016[b] |
| <20 | 10 (3.1) | 0 (0.0) | |
| 20–29 | 60 (18.8) | 2 (3.3) | |
| 30–39 | 63 (19.7) | 5 (7.9) | |
| 40–49 | 61(19.1) | 6 (9.8) | |
| 50–59 | 57 (17.8) | 8 (14.0) | |
| 60+ | 69 (21.6) | 15 (21.7) | |
| **Marital status** | | | <0.001[b] |
| Never married | 68 (21.3) | 3 (4.4) | |
| Married/co-habiting | 14 (4.4) | 15 (8.1) | |
| Separated | 172 (53.8) | 18 (27.3) | |
| **Highest level of education** | | | 0.165[b] |
| None | 64 (20.0) | 10 (15.6) | |
| Primary | 44 (13.8) | 8 (18.2) | |
| JSS/JHS | 120 (37.5) | 11 (9.2) | |
| SSS and above | 92 (28.8) | 7 (7.6) | |
| **#Dietary salt intake** | | | 0.229[b] |
| Often intake | 309 (96.6) | 36 (11.7) | |
| Not often intake | 11 (3.4) | 0 (0.0) | |
| **Alcohol intake** | | | 0.037[b] |
| Yes | 66 (25.2) | 2 (3.03) | |
| No | 196 (74.8) | 23 (11.7) | |
| **Anthropometric parameters** | | | <0.001[a] |
| BMI (Kg/m$^2$) | 23.2±5.1 | 26.4±5.8 | |
| Normal (<25 Kg/m$^2$) | 228 (71.3) | 17 (7.5) | |
| Overweight (25–29 Kg/m$^2$) | 58 (18.3) | 8 (13.8 | |
| Obese (≥30 Kg/m$^2$) | 34 (10.6) | 11 (32.4) | |
| BSA (m$^2$) | 1.6±0.3 | 1.7±0.3 | 0.181[c] |

The values are presented as number (%) or mean ± SD, Chi-square test ($\chi^2$)[a], Fisher's exact[b], t-test[c], # Dietary salt intake: "often intake" (i.e. at least 5g or one teaspoon of iodized salt per day) and "not often intake" (i.e. less than 5g or one teaspoon of iodized salt per day or not at all).

## Discussion

This study sought to contribute to knowledge on thyroid disorders in Ghana using ultrasound imaging to examine the thyroid gland of a sample drawn from six communities in the Assin North District of the Central region of Ghana. The study specifically estimated the prevalence, characterized and determined factors associated with ATN. Prior studies on ATN from a variety of contexts report prevalence ranging from 15 to 67% [11, 12, 15, 24–27]. In the current study the prevalence of ATN was found to be approximately 11%, which is far lower than minimum reported in prior studies. This finding could be an indication that Ghana's iodization educational campaigns which began in the 1990's is finally beginning to yield some benefits. Indeed, an overwhelming majority (97%) of the sample in this study reported that they often consume dietary salt. Much as the study did not ascertain the iodine content participants

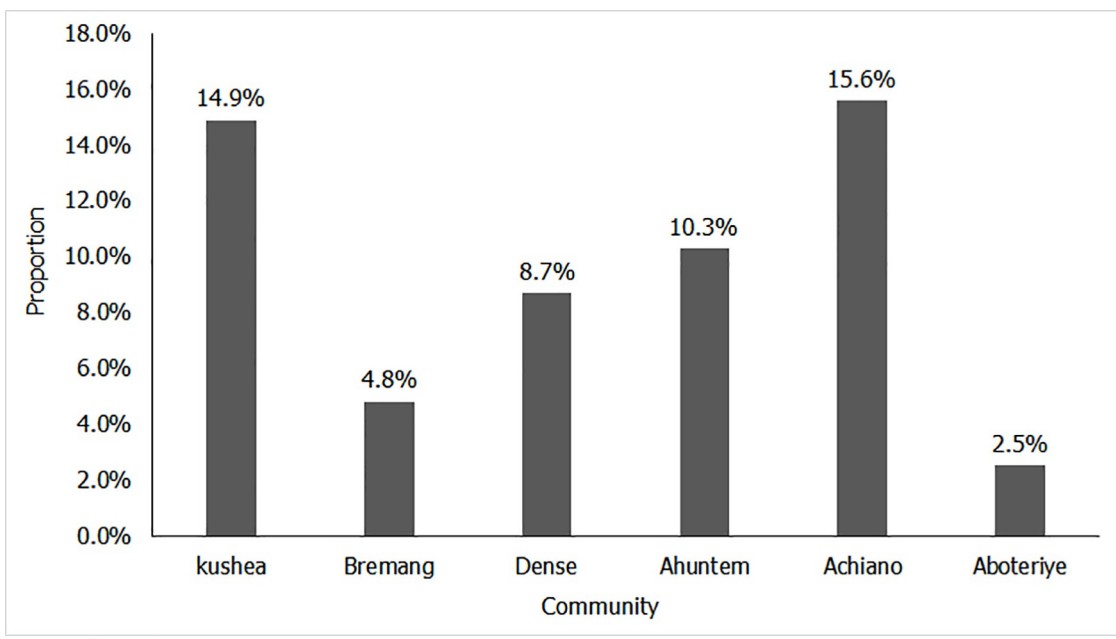

**Fig 1. Distribution of asymptomatic thyroid nodules in sampled population by community.**

Table 3. Ultrasound characteristic of thyroid nodules of study population.

| Characteristics | Subjects with nodules | |
|---|---|---|
| | **Number (n = 36)** | **Percentage (%)** |
| **Location of nodules** | | |
| Left lobe | 14 | 38.9 |
| Right lobe | 9 | 25.0 |
| Both lobes | 7 | 19.4 |
| Isthmus | 2 | 5.6 |
| Both lobes + Isthmus | 4 | 11.1 |
| **Number of nodules** | | |
| Solitary nodule | 23 | 63.9 |
| Multiple nodules | 13 | 36.1 |
| **Nodule Composition** | | |
| Cyst | 2 | 5.6 |
| Solid | 34 | 94.4 |
| **Calcifications** | | |
| No calcification | 35 | 97.2 |
| Calcification | 1 | 2.8 |
| **Dimension of nodules (n = 48)** | | |
| Length/diameter (mean ± sd) | 1.2±0.5 | |
| <0.5 cm | 1 | 2.1 |
| 0.5–1.0 cm | 19 | 39.6 |
| >1.0 cm | 28 | 58.3 |
| **Width (cm) (mean ± sd)** | 0.9±0.5 | |
| <0.5 cm | 4 | 8.3 |
| 0.5–1.0 cm | 32 | 66.7 |
| >1.0 cm | 12 | 25.0 |

**Table 4. Logistic regression analysis of selected variables and presence of nodules.**

| Variable | Unadjusted analysis | | Adjusted analysis | |
|---|---|---|---|---|
| | cOR (95%CI) | p-value | aOR (95%CI) | p-value |
| **Demographic characteristic** | | | | |
| **Sex** | | | | |
| Male | 1 (ref) | | 1 (ref) | |
| Female | 4.54 (1.71–12.01) | 0.002 | 2.32 (0.40–13.49) | 0.350 |
| **Age** | | | | |
| <20 | - | - | - | - |
| 20–29 | 1 (ref) | | 1 (ref) | |
| 30–39 | 2.49 (0.47–13.41) | 0.285 | 10.64 (0.70–161.66) | 0.089 |
| 40–49 | 3.16 (0.61–16.35) | 0.169 | 8.76 (0.54–141.80) | 0.127 |
| 50–59 | 4.73 (0.96–23.35) | 0.056 | 5.00 (0.22–11.31) | 0.309 |
| 60+ | 8.06 (1.76–36.88) | 0.007 | 19.05 (0.99–366.08) | 0.051 |
| **Marital status** | | | | |
| Never married | 1 (ref) | | 1 (ref) | |
| Married/co-habiting | 1.90 (0.53–6.78) | 0.322 | 0.24 (0.02–2.38) | 0.225 |
| Separated | 8.13 (2.26–29.16) | 0.001 | 1.04 (0.09–12.51) | 0.973 |
| **Highest level of education** | | | | |
| No formal education | 1 (ref) | | 1 (ref) | |
| Primary | 1.20 (0.43–3.33) | 0.726 | 1.27 (0.29–5.49) | 0.754 |
| JSS/JHS | 0.54 (0.22–1.36) | 0.194 | 0.63 (0.18–2.20) | 0.472 |
| SSS and above | 0.44 (0.16–1.24) | 0.121 | 0.55 (0.10–3.03) | 0.491 |
| **Dietary salt intake** | | | | |
| Often | N/A | N/A | N/A | N/A |
| Not often | N/A | N/A | N/A | N/A |
| **Alcohol intake** | | | | |
| No | 1 (ref) | | 1 (ref) | |
| Yes | 0.24 (0.05–1.03) | 0.054 | 0.25 (0.04–1.49) | 0.127 |
| **BMI** | | | | |
| Normal (<25 Kg/m$^2$) | 1 (ref) | | 1 (ref) | |
| Overweight (25–29 Kg/m$^2$) | 1.99 (0.81–4.86) | 0.133 | 5.87 (1.11–31.13) | 0.038* |
| Obese (≥30 Kg/m$^2$) | 5.94 (2.48–14.20) | <0.001 | 13.60 (1.46–126.83) | 0.022* |
| BSA (m$^2$) | 1.99 (0.72–5.51) | 0.182 | 0.47 (0.06–3.90) | 0.482 |

cOR- crude odds ratios, aOR- adjusted odds ratios, CI- Confidence interval

* significant at p-value<0.05, N/A-Not applicable, no nodule was recoded in not often dietary salt intake group.

consumed, chances are that iodized dietary salt would be of choice as a result of educational campaign.

Regarding the various characteristic of ATN, the findings in the study both concur and contrast previous studies. In terms of location, Moifo et al. [21] found most ATN in the right lobe, in contrast to the present study where ATN were predominantly located in the left lobe. These variations in location could be the result of native sizes differences between right and left lobes [21]. On ultrasound features examination, our finding that solitary (64%) ATN were more common than multinodular (36%) nodules is consistent with the findings of Kamran et al. [25], but contradicts Shayeb et al. [2] who found multinodular (59%) thyroid to be more common than solitary (41%) ones [2]. Similar to Shayeb et al. [2], but contrary to others [10, 12],

we found a very small proportion of cystic ATN (6%) compared with solid nodules (94%). However, data from this study could not explain the observed variation.

Calcification does not seem to be a common feature in ATN as noted in other studies and confirmed by our finding of a single case with calcification in the current study [2, 12]. We found a higher proportion of nodules greater than 10mm, which resonated with the findings of Kim et al. [13] although the absolute proportions (58% in our study vs 67% in Kim et al.) vary slightly. However, Moifa et al. [21] and Kamran et al. [25] reported much lower proportions (22% and 43% respectively) of nodules greater than 10mm.

The unwavering effect of age on a number of health-related conditions was yet again manifested in this study with the prevalence of ATN significantly increasing with age. Beyond this general pattern of association between age and the development of ATN, our regression analysis showed that those ages 60 years and above have significantly higher probability of developing ATN compared with any other younger age groups. This finding corroborates previous studies and can be linked with the physiologic process of ageing of the thyroid gland [2, 11, 13, 15, 17, 21, 25, 28, 29].

Women seem to disproportionately develop ATN compared with men as found previously and reaffirmed by our current study [2, 10–12, 21, 25–29, 30]. We found the ratio of developing ATN between men and women to 1:6.6. Although not significant at the multivariate model, general positive association between women and the development of ATN was sustained. This could be partly be linked with the reproductive health-related factors of women, particularly pregnancy, child birth, routine changes in menstrual cycles, and menopause [15, 16, 31].

Another important predictor of one developing ATN demonstrated in this study is BMI. Although previous study [10] similarly associated higher BMI with higher prevalence of ATN, this study further demonstrated that being in the overweight or obese category increased the probability of developing ATN by over five and thirteen folds, respectively. The mechanism linking BMI and ATN is not too clear, however, it is suggested that obesity and insulin resistance increase thyroid stimulating hormone secretion via leptin signaling, ultimately resulting in the expansion of thyroid volume and formation of nodules [10]. The effects of educational level and marital status on ATN do not persist beyond the bivariate model in this study, although the results generally point to the protection that being educated or married offers with respect to health [32, 33].

### Limitations of the study

The two main limitations of this study were inadequate measurement of iodize salt intake and under representation of data. Our measurements of ionized salt consumption were inadequate as only one question with binary response was used. Also, our data is an underrepresentation of the Ghanaian population as only one district out of 275 districts and one region out of sixteen regions in Ghana were involved. This study did not cover the entire region or Ghana due to resource constraints. However, our objectives of providing preliminary local community data estimating the prevalence of and characterized asymptomatic thyroid nodule to influence stakeholder discussion and further assessments has been achieved.

### Conclusion

The prevalence of ATN in our study population was lower than previously reported in both developed and developing countries. Solitary nodules were predominant and mostly found in the left lobe of thyroid gland. The key predictors of ATN are age 60 years or older, overweight and obesity. These findings call for the reinforcement and intensification of educational

campaigns on the consumption of iodized dietary salt as well as the consumption of foods rich in iodine content, particularly among older persons. Such educational efforts also include messages on the importance of maintaining a healthy weight.

## Acknowledgments

We acknowledge the assistance given to us by Level 400 medical students (2018/2019 batch) of the School of Medical Sciences, University of Cape Coast, for administering questionnaires to study participants and measuring anthropometric parameters. We would also like to thank residents of Bremang, Dense, Aboteriyie, Ahuntem, Achiano, and Kushea in the Assin North District for their participation.

## Author Contributions

**Conceptualization:** Martin Tangnaa Morna, Derek Anamaale Tuoyire, Ganiyu Adebisi Rahman.

**Data curation:** Bashiru Babatunde Jimah, Anthony Baffour Appiah.

**Formal analysis:** Anthony Baffour Appiah.

**Investigation:** Bashiru Babatunde Jimah, Sebastian Eliason, Anthony Baffour Appiah.

**Methodology:** Derek Anamaale Tuoyire, Ganiyu Adebisi Rahman.

**Resources:** Martin Tangnaa Morna.

**Supervision:** Martin Tangnaa Morna, Sebastian Eliason, Ganiyu Adebisi Rahman.

**Validation:** Bashiru Babatunde Jimah.

**Writing – original draft:** Derek Anamaale Tuoyire, Anthony Baffour Appiah.

**Writing – review & editing:** Martin Tangnaa Morna, Derek Anamaale Tuoyire, Bashiru Babatunde Jimah, Sebastian Eliason, Anthony Baffour Appiah, Ganiyu Adebisi Rahman.

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
