## [Decision Letter · Decision Letter 0]

28 Jan 2021

PONE-D-20-33859

Prevalence and Characterization of Asymptomatic Thyroid Nodules in Assin North District, Ghana

PLOS ONE

Dear Dr. Morna,

Thank you for submitting your manuscript to PLOS ONE. After careful consideration, we feel that it has merit but does not fully meet PLOS ONE’s publication criteria as it currently stands. Therefore, we invite you to submit a revised version of the manuscript that addresses the points raised during the review process.

We look forward to receiving your revised manuscript.

Kind regards,

Francis Moore, Jr.

Academic Editor

PLOS ONE

Additional Editor Comments:

A major revision is required of this interesting study. Please see the comments of Reviewer #1

Journal Requirements:

2.) Please amend your Ethics Statement to include information about informed patient consent provided for the study. Please state what type of consent was given (i.e., written, verbal, etc.). Additionally, please speciify the type of consent in the Methods section of the manuscript.

3.) We note that you have indicated that data from this study are available upon request. PLOS only allows data to be available upon request if there are legal or ethical restrictions on sharing data publicly. For information on unacceptable data access restrictions, please see http://journals.plos.org/plosone/s/data-availability#loc-unacceptable-data-access-restrictions.

Reviewers' comments:

Reviewer's Responses to Questions

**Comments to the Author**

1. Is the manuscript technically sound, and do the data support the conclusions?

Reviewer #1: Yes

2. Has the statistical analysis been performed appropriately and rigorously? 

Reviewer #1: No

3. Have the authors made all data underlying the findings in their manuscript fully available?

Reviewer #1: No

4. Is the manuscript presented in an intelligible fashion and written in standard English?

Reviewer #1: Yes

5. Review Comments to the Author

Reviewer #1: This paper presents a study on the prevalence of Thyroid Nodules in a region of Ghana.

The study is interesting, but the paper requires some more details on the sampling and data collection of the study, as well as on the methods that were used to adjust the estimates of prevalence. The paper would benefit from some editing for clarity throughout.

Major comments:

Introduction: A description about the country or city in terms of previous estimates or importance of estimating thyroid nodule is required.

Methods:

1. The authors mentioned that the household were selected using systematic random sampling. but sampling method is inadequate. Sampling frame should be clearly defined. How were the household defined and selected? What list did you select from?

2. One of the most important question in this study was the salt intake. However, there is no information regarding the items of this scale. in the result, there are just two answer (often, not often) for this question that is not enough. more description about the measurement of this question is required.

3. The statistical method for estimating the prevalence is not satisfactory. A more valid weighting estimate according to the population size of each community such as inverse probability weighting is required.

Results: According to the previous comment on calculation a weighted estimate of prevalence, the result need to be changed and corrected.

Minor comment:

Throughout, the language in the manuscript could benefit from editing by a native English speaker. The authors’ meaning is clear (nearly) everywhere, but the manuscript could benefit from a careful editorial review. Some words are unclear such as; contests, chin, multivariate level (instead of multivariate model), associating, ….

6. PLOS authors have the option to publish the peer review history of their article (what does this mean?). If published, this will include your full peer review and any attached files.

Reviewer #1: No

---

## [Author Response · Author response to Decision Letter 0]

30 Mar 2021

Response to major comments:

Introduction: 

A description about the country or city in terms of previous estimates or importance of estimating thyroid nodule is required.

Answer #1:

Thank you for your suggestions. We have revised the final paragraph of the introduction with further details on the significance of the study to our country or city and the relevant of estimating thyroid nodules in terms of epidemiological and clinical needs. 

Methods:

1. The authors mentioned that the household were selected using systematic random sampling. but sampling method is inadequate. Sampling frame should be clearly defined. How were the household defined and selected? What list did you select from?

Answer #2:

Thank you for your observation and comment. We have revised this section in the manuscript with a clearer definition of what constituted a household in the current study as well as the sampling frame.

2. One of the most important question in this study was the salt intake. However, there is no information regarding the items of this scale. in the result, there are just two answer (often, not often) for this question that is not enough. more description about the measurement of this question is required.

Answer #3:

Thank you for your observation and comment. Your observation is genuine one and we have taken note of that in our study limitations, which should be considered in subsequent studies.

In this study, “often” intake of dietary salt was measured as daily consumption of >5g or one teaspoon of iodized salt, and “not often” intake of dietary salt was measured as daily consumption (<5g or less than one teaspoon of iodized salt or not at all).

All these updates have been made in the main write-up

3. The statistical method for estimating the prevalence is not satisfactory. A more valid weighting estimate according to the population size of each community such as inverse probability weighting is required.

Answer #4:

Thank you for your suggestions and comment. Weighting for dataset in estimating the prevalence of thyroid during analysis was not required as the design addressed sample weighting before data collection stage by taking into consideration sampling proportionate-to-population size in each community 

Results: According to the previous comment on calculation a weighted estimate of prevalence, the result need to be changed and corrected.

Answer #5:

Thank you for your comment and suggestions. We duly acknowledge your genuine comments but we believe this has been duly addressed based on our initial responses in #4 above However, we have added the distribution of symptomatic thyroid nodules by community studied as Figure 1.

Minor comment:

Throughout, the language in the manuscript could benefit from editing by a native English speaker. The authors’ meaning is clear (nearly) everywhere, but the manuscript could benefit from a careful editorial review. Some words are unclear such as; contests, chin, multivariate level (instead of multivariate model), associating,…

Answer #6:

Thank you for your observations and suggestions. We have conducted thorough editing as suggested to make the write-up read better

---

## [Decision Letter · Decision Letter 1]

8 Jun 2021

PONE-D-20-33859R1

Prevalence and Characterization of Asymptomatic Thyroid Nodules in Assin North District, Ghana

PLOS ONE

Dear Dr. Morna,

Thank you for submitting your manuscript to PLOS ONE. After careful consideration, we feel that it has merit but does not fully meet PLOS ONE’s publication criteria as it currently stands. Therefore, we invite you to submit a revised version of the manuscript that addresses the points raised during the review process.

We look forward to receiving your revised manuscript.

Kind regards,

Francis Moore, Jr.

Academic Editor

PLOS ONE

Journal Requirements:

Additional Editor Comments (if provided):

Please check the references.

For instance, #27 was cited earlier as #21.

Reviewers' comments:

Reviewer's Responses to Questions

**Comments to the Author**

1. If the authors have adequately addressed your comments raised in a previous round of review and you feel that this manuscript is now acceptable for publication, you may indicate that here to bypass the “Comments to the Author” section, enter your conflict of interest statement in the “Confidential to Editor” section, and submit your "Accept" recommendation.

Reviewer #2: All comments have been addressed

2. Is the manuscript technically sound, and do the data support the conclusions?

Reviewer #2: Yes

3. Has the statistical analysis been performed appropriately and rigorously? 

Reviewer #2: Yes

4. Have the authors made all data underlying the findings in their manuscript fully available?

Reviewer #2: Yes

5. Is the manuscript presented in an intelligible fashion and written in standard English?

Reviewer #2: Yes

6. Review Comments to the Author

Reviewer #2: The manuscript is well written and all the necessary requirements complied with. The ethical and publication ethics have been complied with.

7. PLOS authors have the option to publish the peer review history of their article (what does this mean?). If published, this will include your full peer review and any attached files.

Reviewer #2: **Yes: **Osei Sarfo-Kantanka

---

## [Author Response · Author response to Decision Letter 1]

12 Jan 2022

Reviewers' comments:

Reviewer's Responses to Questions

Authors’ response: 

In line with the reviewers’ responses to question 1 to 7, no further 

editing was required from the authors. However, we have thoroughly proofread the manuscript to 

correct any minor errors found. Please, see the revised manuscript with Track Changes

---

## [Editor Report · Decision Letter 2]

19 Jan 2022

Prevalence and Characterization of Asymptomatic Thyroid Nodules in Assin North District, Ghana

PONE-D-20-33859R2

Dear Dr. Morna,

We’re pleased to inform you that your manuscript has been judged scientifically suitable for publication and will be formally accepted for publication once it meets all outstanding technical requirements.

Kind regards,

Francis Moore, Jr.

Academic Editor

PLOS ONE

---

## [Editor Report · Acceptance letter]

24 Jan 2022

PONE-D-20-33859R2 

Prevalence and Characterization of Asymptomatic Thyroid Nodules in Assin North District, Ghana 

Dear Dr. Morna:

I'm pleased to inform you that your manuscript has been deemed suitable for publication in PLOS ONE. Congratulations! Your manuscript is now with our production department. 

Kind regards, 

on behalf of

Dr. Francis Moore, Jr. 

Academic Editor

PLOS ONE